# Identification of a New Heavy-Metal-Resistant Strain of *Geobacillus stearothermophilus* Isolated from a Hydrothermally Active Volcanic Area in Southern Italy

**DOI:** 10.3390/ijerph17082678

**Published:** 2020-04-14

**Authors:** Rosanna Puopolo, Giovanni Gallo, Angela Mormone, Danila Limauro, Patrizia Contursi, Monica Piochi, Simonetta Bartolucci, Gabriella Fiorentino

**Affiliations:** 1Dipartimento di Biologia, Università Degli Studi di Napoli Federico II, 80139 Napoli, Italy; rosanna.puopolo@unina.it (R.P.); giovanni.gallo2@unina.it (G.G.); limauro@unina.it (D.L.); contursi@unina.it (P.C.); bartoluc@unina.it (S.B.); 2Istituto Nazionale di Geofisica e Vulcanologia, Sezione Osservatorio Vesuviano, 80125 Napoli, Italy; angela.mormone@ingv.it (A.M.); monica.piochi@ingv.it (M.P.)

**Keywords:** heavy metals, thermophiles, *Geobacillus stearothermophilus*, minimal inhibitory concentration (MIC), transmission electron microscopy (TEM)

## Abstract

Microorganisms thriving in hot springs and hydrothermally active volcanic areas are dynamically involved in heavy-metal biogeochemical cycles; they have developed peculiar resistance systems to cope with such metals which nowadays can be considered among the most permanent and toxic pollutants for humans and the environment. For this reason, their exploitation is functional to unravel mechanisms of toxic-metal detoxification and to address bioremediation of heavy-metal pollution with eco-sustainable approaches. In this work, we isolated a novel strain of the thermophilic bacterium *Geobacillus stearothermophilus* from the solfataric mud pool in Pisciarelli, a well-known hydrothermally active zone of the Campi Flegrei volcano located near Naples in Italy, and characterized it by ribotyping, 16S rRNA sequencing and mass spectrometry analyses. The minimal inhibitory concentration (MIC) toward several heavy-metal ions indicated that the novel *G. stearothermophilus* isolate is particularly resistant to some of them. Functional and morphological analyses suggest that it is endowed with metal resistance systems for arsenic and cadmium detoxification.

## 1. Introduction

Heavy metals are among the most persistent and toxic pollutants. Differently from organic xenobiotics, which can be converted into non-harmful substances, they cannot be completely removed from the environment [1]. Anthropogenic sources including mining and various industrial (vehicle emissions, lead–acid batteries, paints) and agricultural activities (fertilizers, aging water supply) lead to their increasing accumulation [2,3,4]. The prolonged exposure to heavy metals is responsible for several human diseases, as documented by the World Health Organization [5]. For example, arsenic and cadmium have been classified as carcinogenic compounds by the International Agency for Research on Cancer (IARC) in both humans and animals [6,7], while the exposure to lead is responsible for 3% of cerebrovascular disease worldwide [8]. Therefore, reduction of heavy-metal pollution is actually among the greatest challenges of the new century [9,10,11]. 

As a consequence of the massive accumulation of toxic metals into the environment, majority of microorganisms have developed genetic resistance mechanisms [12,13,14] and even specific metabolic pathways to deal with toxic metals [15,16,17]. For instance, gram-positive and gram-negative bacteria possess arsenic resistance systems encoded by operons carried either on plasmids or on the chromosome. Genetic determinants are usually the three genes *arsR, arsB*, and *arsC* [18,19], among which *arsR* encodes a trans-acting repressor of the ArsR/SmtB family involved in transcriptional regulation [20,21,22], *arsB* encodes an As(III) efflux transporter (ArsB/Acr3) [23], and *arsC* encodes a cytoplasmic arsenate reductase that converts As(V) to As(III), the latter extruded outside the cell [24,25,26]. Other organisms benefit of additional proteins that improve the arsenic resistance, such as the arsenite methyltransferases [27]. The arsenic resistance system of some microorganisms also provides Cd(II) tolerance [28,29,30]. Gram-positive and gram-negative bacteria also possess cadmium resistance systems, which are generally composed by two genes, *cadC* coding for a helix-turn-helix transcriptional regulator that controls the second gene *cadA*, coding for a cadmium-translocating P-type ATPase. The loci of genes involved in cadmium resistance are either on plasmids or on chromosomes [31].

Since many metals, such as arsenic, are naturally present in volcanic and geothermal springs, these niches are commonly colonized by heavy-metal-resistant microorganisms [32,33]; they actively participate in geochemical cycles, solubilizing and precipitating metals thus contributing to transforming the bedrock and remodeling their ecosystems [34]. The interest in the comprehension of the molecular mechanisms underlying heavy-metal resistance in microorganisms thriving in extreme environments is growing fast because of the urgent need to develop effective and eco-sustainable approaches toward heavy-metal pollution [35,36,37,38].

In this work, we describe the isolation and characterization of a new thermophilic heavy-metal-resistant microorganism from the solfataric mud pool of Pisciarelli in the Campi Flegrei volcano located near Naples in Italy. The site has extreme environmental conditions in terms of temperature, acidity, and As-rich geochemistry due to an intense hydrothermal activity [39,40].

## 2. Materials and Methods

### 2.1. Chemicals

The antibiotics and metal salts used in this work were purchased by Sigma-Aldrich. Antibiotics, identified through the unique number assigned by the Chemical Abstracts Service (CAS), are: Ampicillin (CAS number: 7177-48-2); bleomycin sulfate (CAS number: 9041-93-4); chloramphenicol (CAS number: 56-75-7); kanamycin sulfate (CAS number: 70560-51-9); hygromycin B (CAS number: 31282-04-9); tetracycline (CAS number: 60-54-8). Metal salts used are the following: sodium (meta)arsenite (NaAsO_2_); sodium arsenate dibasic heptahydrate (Na_2_HAsO_4_ · 7H_2_O); cadmium chloride (CdCl_2_); cobalt chloride hexahydrate (CoCl_2_ · 6H_2_O); cobalt chloride (CoCl_3_); chromium oxide (CrO_3_); copper nitrate trihydrate [Cu(NO_3_)_2_ · 3H_2_O]; mercury chloride, (HgCl_2_); nickel chloride (NiCl_2_); sodium orthovanadate dihydrate (Na_3_VO_4_ · 2H_2_O).

### 2.2. Strain Isolation and Molecular Identification

Soil samples were collected in April 2016 at the hydrothermal site of Pisciarelli (Figure 1) that lies on the Solfatara volcano, one of the various eruptive vents generated within the wide volcanic field of the Campi Flegrei since ca. 4100 years [41]. 

The materials were aseptically sampled from the main mud pool; both pH and temperature were measured contextually by indicator papers and a Fluke digital thermocouple probe, respectively. The water was collected from the bubbling mud pool, while the mud was taken from its marginal water-poorer portion (Figure 1). Temperature and pH values were 94.3 °C and 4.8 in the bubbling mud pool, respectively, while in the marginal water-poorer portion they were 55.3 °C and 6.0, respectively. The local levels of arsenic are in the 10–20 ppm range [39].

Enrichment cultures were set in 50 mL tubes with 20 mL of Luria-Bertani (LB) medium [42] and inoculated with 1 g of soil sample at 37, 50, and 75 °C for 24 h. Then serial dilutions of these culture samples were streaked on LB-agar plates (1.5%) at the same temperature values for 24 h. Bacterial colonies were found in the mud sample incubated at 50 °C and the isolation of a pure strain was carried out by repeated streak plating on solid medium prepared with the LB-agar incubated at 50 °C.

To identify the new isolate, we used different approaches based on standard procedures such as automated ribotyping on digested chromosomal DNA and fatty acid analysis [43]. Since the taxonomic identification at the species level could not be determined with ribotyping data, a MALDI-TOF MS analysis (i.e., Matrix-Assisted Laser Desorption/Ionization Mass Spectrometry performed with a Time-Of-Flight Mass Spectrometer) was also commissioned to the Leibniz Institute DSMZ (the German Collection of Microorganisms and Cell Cultures, GmbH). Sample preparation and instrumental conditions have been described elsewhere [44]. The identification report was generated by the Biotyper software (version 3.1, Bruker Daltonics GmbH, Bremen, Germany) and the strain was identified as *Geobacillus stearothermophilus* with a score corresponding to high probable strain identification (higher than 2.3).

Moreover, 16S rRNA sequencing was commissioned to Eurofins. The resulting sequencing data (about 1000 nt) were analyzed in the nucleotide database of the U.S. National Center for Biotechnology Information (NCBI, www.ncbi.nlm.nih.gov). The sequence in the sample could be identified as a novel strain of *Geobacillus stearothermophilus (G. stearothermophilus* GF16 hereafter). It was also submitted to GenBank and the accession number is MT311361.

#### Neighbor-Joining Tree Development

In order to obtain information about evolutionary relationships within *G. stearothermophilus* species, the 16S rRNA sequence of the novel *G. stearothermophilus* GF16 isolate was analyzed with BLASTn; sequences with identities from 99% to 97% were aligned using the multiple sequence alignment tool CLUSTAL Omega (https://www.ebi.ac.uk/Tools/msa/clustalo/). Finally, a neighbor-joining tree was constructed using the default option of ClustalW2 (Simple Phylogeny) [45].

### 2.3. Geobacillus stearothermophilus Physiological Analyses

#### 2.3.1. Determination of Optimal Growth Conditions

A frozen (−80 °C) glycerol-stock of *G. stearothermophilus* GF16 was streaked on an LB-agar plate and incubated overnight at 50 °C. Single colonies that appeared on the plate were inoculated into liquid LB media at different pH values (pH 3, pH 5, pH 7) at 50 and 60 °C under shaking. Growth was only observed at pH 7.0, whereas the optimal growth temperature turned out to be 60 °C. The generation time (*G*) was calculated with the following formula: *G* = *t*/*n*, where *t* is the time interval and *n* the number of generations (*t* is considered to be between 3 and 4 h, in the exponential phase). All the experiments were repeated in triplicate.

#### 2.3.2. Antibiotic Susceptibility

For the determination of minimal inhibitory concentrations (MIC) toward antibiotics, a modified version of a protocol described in the Manual of Antimicrobial Susceptibility Testing was followed [46]. In detail, a frozen (−80 °C) stock of *G. stearothermophilus* GF16 was streaked on an LB-agar plate and incubated at 60 °C overnight. A single colony was inoculated into liquid LB medium and incubated at 60 °C under shaking up to the exponential phase corresponding to Optical Density (OD) at 600 nm of 1.5 (OD_600nm_ were measured in a Varian Cary 50 Scan UV-Visible Spectrophotometer). Then the bacterial culture was diluted up to 0.1 OD_600nm_ in LB medium supplemented with increasing concentrations (from 5 to 50 µg/mL) of antibiotics (ampicillin, kanamycin, chloramphenicol, tetracycline, hygromycin, and bleomycin) and grown at 60 °C for 16 h; for each determination, three independent experiments were carried out in triplicate. Minimum inhibitory concentration (MIC) was determined as the lowest concentration of antibiotics that completely inhibited the growth of the strain as evaluated by OD_600nm_ measurements after incubation for 16 h under optimal growth conditions. 

#### 2.3.3. Heavy-Metal Resistance

For the determination of MIC toward heavy-metal ions (As(V), As(III), Cd(II), Co(III), Cr(VI), Cu(II), Hg(II), Ni(II), V(V)), cell cultures were grown and diluted as described above. The heavy metals were added at increasing concentration ranging from 0.1 to 120 mM. The MIC values were determined as described above. The values reported are the average of three independent experiments each one performed in triplicate. 

#### 2.3.4. Evaluation of As(V) Biotransformation

The As(V) transformation capacity of *G. stearothermophilus* GF16 to produce As(III) was qualitatively evaluated using a colorimetric assay based on the formation of precipitates upon reaction of AgNO_3_ with arsenic [47]. A single colony was cultured in LB liquid medium at 60 °C up to 1.0 OD_600nm_; then an aliquot of the cell suspension was streaked on LB-agar plates containing 50 mM sodium arsenate (Na_2_HAsO_4_). The LB-agar plates were incubated at 60 °C for 18 h and then flooded with 0.1 M AgNO_3_. As controls, the following plates were prepared: (1) LB agar supplemented with As(V), without cells; (2) LB agar, without As(V), with streaked cells; (3) LB agar without either cells or As(V). The color of the precipitate on each plate was compared to a color scale, which could be used as reference to distinguish by eye different ratios of As(V)/As(III), thus allowing a qualitative evaluation of As(III) production. The reference color scale was developed by mixing defined ratios of As(V) and As(III) in different tubes. The final concentration of total arsenic was 50 mM for all the samples. All the experiments were repeated in triplicate.

#### 2.3.5. Bioinformatic Analysis

Bioinformatic analyses were performed to evaluate the presence of arsenic and/or cadmium resistance genes in the genomes of the following three sequenced strains of *G. stearothermophilus*: (a) strain n° 10 (Accession BioProject PRJNA252389); (b) strain DSM 458 (Accession BioProject PRJNA327158); (c) strain B5 (Accession Bioproject PRJNA513473). Loci containing sequences coding for putative arsenic resistance proteins were identified on the NCBI database, and the corresponding translated sequences were aligned with the multiple sequence alignment program Clustal Omega (https://www.ebi.ac.uk/Tools/msa/clustalo/).

#### 2.3.6. Transmission Electron Microscopy (TEM)

To analyze cell morphology, *G. stearothermophilus* GF16 was grown at 60 °C in LB medium (pH 7), and in LB (pH 7) supplemented with As(V) 117 mM or Cd(II) 0.9 mM for 16 h; a control grown in the absence of heavy-metal ions was also harvested at 1.5 OD_600nm_ corresponding to a mid-exponential growth phase. Cells were pelleted by centrifugation, washed twice with phosphate buffered saline (PBS 1%), and fixed as reported by Pinho et al. [48]. Resi-sections were prepared with the ultramicrotome (LKB SuperNova) and serially stained with uranyl acetate and lead citrate. The sections were then studied on a Philips EM 208s Transmission Electron Microscope. Control cells not subjected to metal treatment were compared with heavy-metal-treated samples to check for possible heavy-metal accumulation.

## 3. Results and Discussion

### 3.1. Geochemical Characterization of the Sampling Site

Like similar volcanic systems worldwide, the Solfatara volcano hosts an acidic sulfate environment determined by the hot circulation of aggressive sulfurous fluids deriving from mixing between deeply infiltrating meteoric waters and ascending magmatic gases [39,40,41]. This phenomenon causes intense rock alteration and concentration of certain elements, such as As [39,49,50,51,52,53,54]. However, differently from the diffuse and fumarolic outgassing characterizing the Solfatara crater, the Pisciarelli site is a water-dominant environment, showing the formation of boiling pools and water springs and the opening of low-energetic geyser-type vents. The site represents the shallowest portion of a widespread geothermal system that develops in the subsurface and converts into brines downward to its deeper roots that are directly supplied by the magmatic outgas. Due to the increased hydrothermal activity since 2006, the site shows maximum temperatures of ca. 110 °C and up to 260 tons/day of CO_2_ [55] with an abundance of H_2_S and the presence of minor gaseous species such as CH_4_, N_2_, H_2_, and CO. 

At the time of sampling, the bubbling mud pool was at pH 4.8 and 94.3 °C and the marginal mud at pH 6.0 and 55.3 °C, while surrounding soils were at temperature up to 98–99 °C and very acidic pH. These values are in the range known for the area, although lower temperatures were also measured in the mud pool (approximately 70 °C). Furthermore, the mineralogical and chemical features of the sampled materials [39] are those usually determined. Indeed, the mud was typically gray in color and essentially enclosed sulfates (i.e., K- and Al- bearing alunite), sulfides (i.e., Fe- plus S-bearing pyrite), and silica-phases; dried water samples crystallized NH_4_-bearing sulfates. The mud is enriched in As (10–20 ppm) and Hg (around 40 ppm) compared to the protolith volcanic deposits; contains few wt.% of Fe_2_O_3_; approximately 60 ppm of V; 10–20 ppm of Pb; <10 ppm of Co, Ni, and Cr; 10–20 ppm of Cu; 1–2 ppm of Tl; and practically lacks Cd being at <0.1 ppm [39]. Based on Valentino and Stanzione [56], Pisciarelli waters are rich in SO_4_^-2^ (1400–7000 mg/L) and NH_4_ 500–1000 mg/L); contain F (0.5–30 mg/L), Al (65–20 mg/L), and B (0.1–0.8 mg/L); lack carbonate species and chlorine; the content of As, Hg, Tl, Pb is approximately 40–2000, 40–250, 2–8, and 5–30 µg/L, respectively. The general enrichment in S, NH_4_, As, and Hg is consistent with the volcanic setting and the magmatic/geothermal outgas support. According to Aiuppa et al. [54], arsenate is the As-compost under equilibrium in the water solutions.

### 3.2. Isolation and Identification of Geobacillus stearothermophilus GF16

Upon incubation of mud samples taken from the marginal water-poorer portion, cell growth was observed in LB medium at pH 7 and 50 °C. Single colonies were isolated by serial dilutions in the same medium, and the isolated strain showed an optimal growth temperature of 60 °C.

In order to identify the microorganism, ribotyping and fatty acid analyses were performed at DSMZ; the results led to the identification of a member of *Geobacillus* genus but did not allow to differentiate unambiguously at species level. Geobacilli were first described by Nazina et al. in 2001 [57]; they are gram-positive, endospore-forming, aerobic or facultative anaerobic thermophiles, growing optimally at temperatures between 50 and 72 °C and exploitable for various biotechnological applications such as for bioremediation and production of thermostable enzymes and biofuels [58,59]. The interest toward microorganisms of the *Geobacillus* genus prompted us to combine two different experimental approaches such as MALDI-TOF MS analysis and 16S rRNA sequencing to try to unambiguously identify the species. Indeed, the classification of the different species within the *Geobacillus* genus is challenging since the sequence similarity of the 16S rRNA can be higher than 97% even among species [60]. On the other hand, MALDI-TOF MS analysis has been proposed as a powerful bioanalytical method to detect profiles of proteins derived from whole bacterial cells to be used for bacteria identification [61]. The combined molecular approaches allowed the identification of a new isolate of *Geobacillus stearothermophilus* that we named *G. stearothermophilus* GF16. 

Multiple alignment of 16S rRNA sequence of the novel *G. stearothermophilus* isolate (GF16) with those of other Geobacilli and Bacilli with identities from 99% to 97% was performed to build the phylogenetic tree shown in Figure 2. The results confirmed the difficulty in determining a threshold for defining species within the *Geobacillus* genus and supported the concept that a combination of genotypic and phenotypic characteristics could be not sufficient for describing a new species [60].

### 3.3. Metal Ion Resistance and Antibiotic Susceptibility in G. stearothermophilus GF16

To evaluate the sensitivity and the tolerance of *G. stearothermophilus* GF16, MICs toward different antibiotics and heavy metals were determined. For this purpose, the microorganism was grown in the presence of different heavy metals (see Table 1) and antibiotics (ampicillin, kanamycin, chloramphenicol, tetracycline, hygromycin, bleomycin). *G. stearothermophilus* GF16 was found to be sensitive to all the tested antibiotics, even at the lowest concentration used; to the best of our knowledge, no antibiotic resistance has been previously reported for other *G. stearothermophilus* isolates, although the genome of *G. stearothermophilus* 10 contains a sequence coding for a putative tetracycline MFS (Major Facilitator Superfamily) efflux protein (locus tag: GT50_RS17520). 

Interestingly, *G. stearothermophilus* showed high tolerance to As(V) and V(V), as reported in Table 1. Similar MIC values were also found in other Geobacilli such as *G. stearothermophilus* AGH-02 [62], *G. stearothermophilus* ASR4 [63], or *Geobacillus kaustophilus* [64]. The high resistance to both vanadate and arsenate ions was not surprising considering the similarity in their structures; in addition, the structural similarity of both ions with the phosphate ions suggested that V(V) and As(V) could be taken up by cells through phosphate transport systems [65]. 

As for other aerobic microorganisms [66], arsenic resistance within the *Geobacillus* genus relies on the ability to oxidize arsenite, or to reduce arsenate and extrude the arsenite. In particular, As(III) resistance depend on membrane or periplasmatic oxidase activities [67], while resistance to As(V) mainly involves intracellular reductase activities [68] and membrane transporters for As(III) efflux [69]. Since we measured very low tolerance to As(III) (Table 1) in comparison to the values reported in the literature (1.9 mM versus 10–30 mM) [62,63,64], it can be hypothesized either that our isolate lacks arsenite oxidase activity or that the high sensitivity to As(III) is due to the absence of active transport systems for As(III) efflux. For example, the legume symbiont *Sinorhizobium meliloti* was very tolerant to As(V) but very sensitive to As(III) since it was deficient of As(III) transporter systems [65,70]. 

The new isolate was also found to be Cd(II) tolerant, and in this case the MIC value determined was similar to that measured in other Geobacilli (ranging from 0.4 to 3.2 mM) [71]. For the majority of these microorganisms Cd(II) resistance was ascribed to biosorption, i.e., a phenomenon of metal binding to the microbial cell wall, which does not involve energy consumption [72,73]. Interestingly, the Pisciarelli site is enriched in arsenic and vanadium but lacks cadmium (see Section 3.1). Therefore, the presence of genetic determinants for Cd(II) tolerance cannot be traced back to the selective pressure exerted by the environment.

Figure 3 shows the effect of As(V) on *G. stearothermophilus* GF16 growth: the generation time shifted from 30 min for cells grown in the absence of As(V) to 60 and 125 min for those grown in the presence of As(V) 25 and 50 mM, respectively.

As we only observed a high As(V) resistance, we sought to evaluate whether *G. stearothermophilus* GF16 had any As(V) reductase activity; for this purpose, an AgNO_3_ colorimetric method [47,74,75,76] was employed on cells grown on LB-agar supplemented with As(V), using as controls plates of: (1) LB-agar with As(V) and no cells; (2) LB-agar without As(V) and with grown cells; (3) LB-agar without cells and As(V) (Figure 4). The method is based on the formation of colored precipitates upon reaction of AgNO_3_ with arsenic; in particular, the addition of AgNO_3_ to the grown cells produces a brown precipitate (Ag_3_AsO_4_) if AgNO_3_ reacts with As(V) and a bright yellow precipitate (Ag_3_AsO_3_) if AgNO_3_ reacts with As(III) [47,74,75,76]. Therefore, the addition of As(V) to the growth medium implies that As(III) can be revealed only if it is produced inside the cell and extruded afterward. Moreover, as can be seen from the reference color scale in Figure 4E, the solution is clearly yellow only when As(III) is more than 50% of the total arsenic. Figure 4A shows a brown precipitate, indicating that As(V) was the predominant chemical species outside the cells. The negligible amount of extracellular As(III) detected within this experiment, suggested either that *G. stearothermophilus* GF16 had low As(V) reductase activity or could not efficiently extrude As(III). This latter hypothesis might be consistent with the lack or low activity of As(III) efflux systems. However, to confirm these hypotheses, more sensitive experimental approaches such as Inductively Coupled Plasma Mass Spectrometry (ICP-MS), able to detect lower amounts of As(III), are required.

### 3.4. Bionformatic Analyses

To the best of our knowledge, the genomes of only three strains of *G. stearothermophilus* GF16 have been fully sequenced (Table 2): (1) strain “10” isolated from the Yellowstone hot spring (USA); (2) strain “DSM458” isolated from a sugar beet factory in Austria [77]; and (3) strain “B5” isolated from a rice stack in China. As shown in the phylogenetic tree, these strains are evolutionarily very closely related to the GF16 isolate (Figure 2).

In order to verify whether different strains of *G. stearothermophilus* had arsenic and cadmium resistance systems and to understand whether such systems were conserved among the species, a comparative genomic analysis was carried out on the sequences of three *G. stearothermophilus* genomes available at NCBI (Table 2). The study revealed differences in the abundance and type of putative arsenic and cadmium resistance genes in the genomes analyzed (Table 3). In particular, all of them contained one conserved copy of *cadC* and *cadA*: the alignment of the corresponding proteins from the three different strains showed a high degree of identity (92%). This result could explain the tolerance of the new isolate toward Cd(II) despite the absence of this metal within its specific environment (see Section 3.1). Regarding arsenic resistance systems, a copy of ArsB/Acr3 arsenite efflux transporters was found in each genome, whereas at least a simple *ars* system encoding the arsenate reductase (*arsC*) in tandem with an ArsR/SmtB transcriptional regulator was found only in the genomes of *G. stearothermophilus* 10 and B5 strains (86% of identity of both proteins). On the other hand, the strain DSM458 encoded a unique arsenate reductase; moreover, sequences coding for putative arsenite oxidases were not observed in any of the genomes analyzed [78,79]. To the best of our knowledge there are no reports of functional studies on metal resistance systems in these three strains, therefore we conclude that additional investigation is required to shed light on the occurrence of common metal resistance mechanisms in *G. stearothermophilus* isolates. However, the in silico analysis of the genomes showed that the number and type of genes coding for elements involved in arsenic resistance is variable within the same species and depends on the specific evolutionary adaptation of that particular strain [80].

### 3.5. Analysis of Cellular Morphology

In order to better define *G. stearothermophilus* GF16 morphology, we resolved to analyze cells through TEM. As shown in Figure 5, cells have a typical bacillar rod shape when they are actively growing.

Moreover, with the aim of verifying whether As(V) and Cd(II) had any effect on cell morphology, TEM images were also acquired on samples of *G. stearothermophilus* GF16 grown for 16 h in the presence of As(V) and Cd(II) at concentrations corresponding to the MIC values, and they were compared to images of control cells not subjected to any treatment with heavy metal (Figure 6). The sections obtained revealed the structure of the cell more clearly in the control cells (Figure 6A) than in those treated with heavy metals. However, the presence of several cells in division suggested that both As(V) and Cd(II) did not cause significant changes in the cellular structure and cell viability (of Figure 6B vs. Figure 6C). Nevertheless, it appeared that the cell wall of *G. stearothermophilus* GF16 was influenced by handling both As and Cd. In particular, the cell wall of *G. stearothermophilus* GF16 treated with As(V) (Figure 6B) exhibited abundance of ridges and grooves that can be related to a reduction in cell permeability. Interestingly, Cd(II)-treated cells (Figure 6C) appeared darker; this phenomenon could be ascribed to the ability of *G. stearothermophilus* to adsorb Cd(II), as also reported by Hetzer et al. [71]. 

In conclusion, electron microscopy analyses highlighted that the cell shape/structure of *G. stearothermophilus* GF16 changes in presence of As(V) and Cd(II), thus suppling a morphological explanation for the tolerance of the new isolate toward these metal ions.

## 4. Conclusions

With the aim of characterizing new thermophilic heavy-metal-resistant microorganisms, soil sampling was performed in a hydrothermal volcanic area near Naples in Italy, known as Pisciarelli. This is an acidic sulfate area located close to the Solfatara crater famous for an intense endogenous diffuse and fumarolic water-dominant outgassing activity; the chemical composition of mud and water samples revealed that the main metal is iron, but arsenate is an additional significant component. Since geothermal sites are very interesting sources of thermophilic organisms and Pisciarelli is an arsenic-rich area, we hypothesized that novel thermophiles could be found able to detoxify this metal or use it for energy-yielding reactions. We succeeded in isolating a microorganism with an optimal growth temperature of 60 °C and an optimal pH 7, from a water-poor mud. Subsequent molecular identification revealed homology to the species *G. stearothermophilus*. Our laboratory culturing experiments demonstrated the ability of *G. stearothermophilus* GF16 to grow in the presence of arsenate in a range of concentrations comparable to those of bacteria classified as arsenic resistant and in agreement with the natural environmental setting composition as well. This study highlights the adaptation capabilities of the new isolate of *G. stearothermophilus* and its tolerance to extreme environmental conditions and points out to further molecular and physiological investigations to clarify its role in the biogeochemical cycle of arsenic as well as its potential for the management of heavy-metal environmental contaminations.

## Figures and Tables

**Figure 1 ijerph-17-02678-f001:**
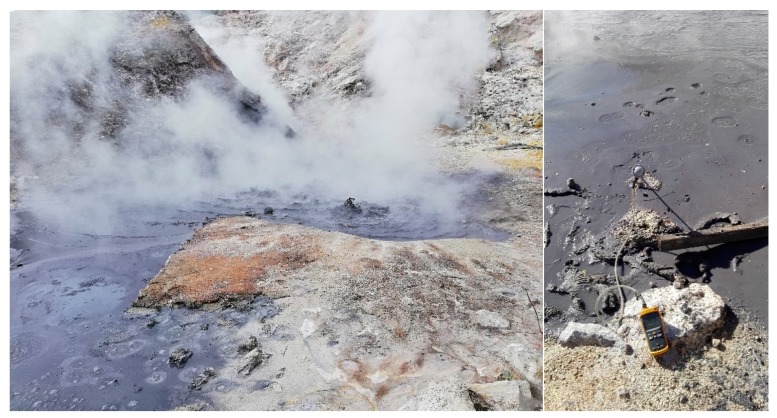
Pisciarelli sampling site showing an intense hydrothermal activity and the puddle water (left) and mud (right) collected.

**Figure 2 ijerph-17-02678-f002:**
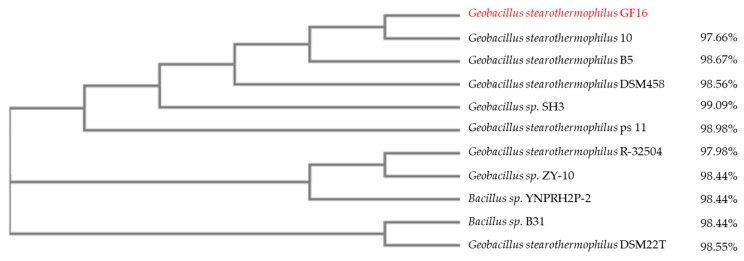
Schematic representation of a phylogenetic tree based on 16S rRNA sequences of different *Geobacillus stearothermophilus* strains. This is a neighbor-joining tree without distance corrections. The new isolate is highlighted in red. The sequence identity (%) to *G. stearothermophilus* GF16 is reported on the right.

**Figure 3 ijerph-17-02678-f003:**
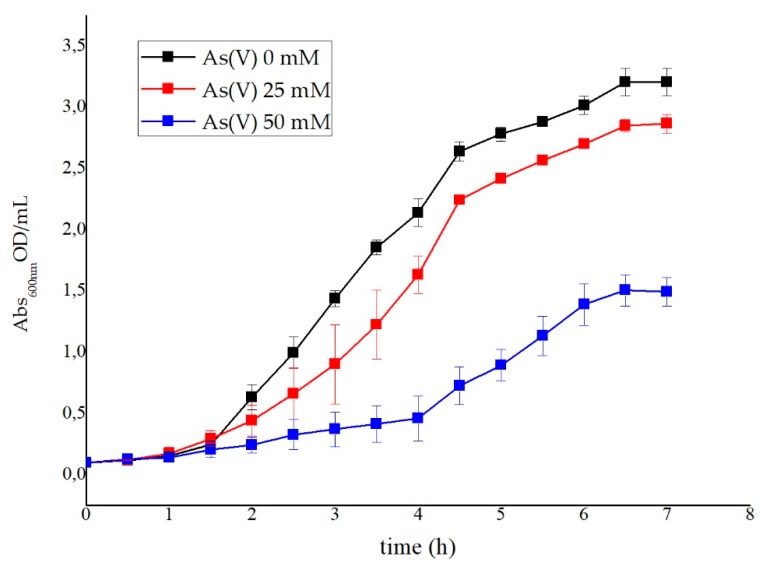
*G. stearothermophilus* GF16 grown in the presence and absence of As(V).

**Figure 4 ijerph-17-02678-f004:**
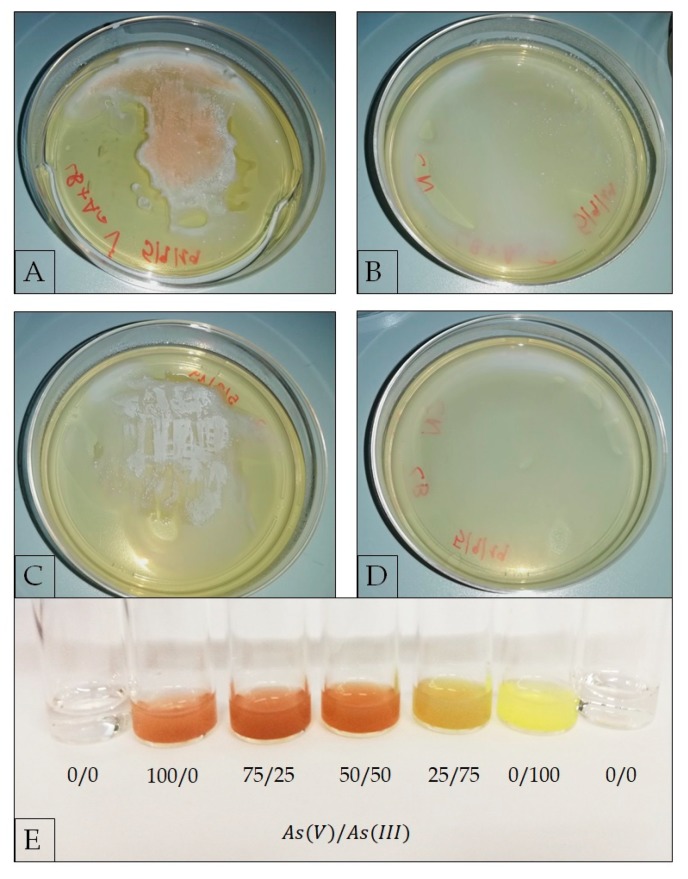
Silver nitrate test on: (**A**) *G. stearothermophilus* GF16 grown on LB-agar plate supplemented with 50 mM As(V); (**B**) LB-agar plate supplemented with 50 mM As(V) (control 1); (**C**) *G. stearothermophilus* grown on LB-agar plate (control 2); (**D**) LB-agar plate (control 3). (**E**) Determination of precipitate color as function of As(V)/As(III) ratio (%). The concentration of total arsenic in solution in each sample is 50 mM (i.e., for the ratio 50/50 there are in solution As(V) 25 mM and As(III) 25 mM).

**Figure 5 ijerph-17-02678-f005:**
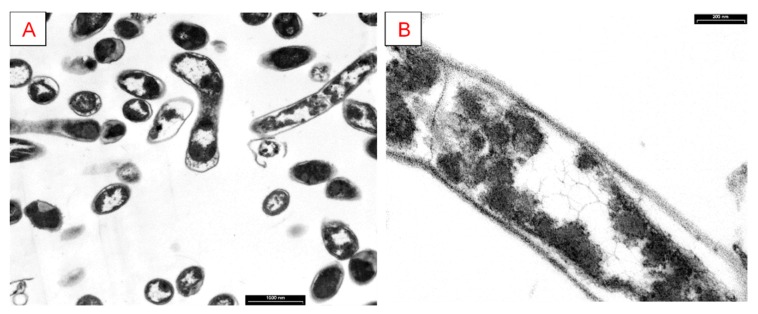
TEM images of *G. stearothermophilus* GF16 in exponential phase at different scales (black bars): (**A**) 1000 nm; (**B**) 200 nm.

**Figure 6 ijerph-17-02678-f006:**
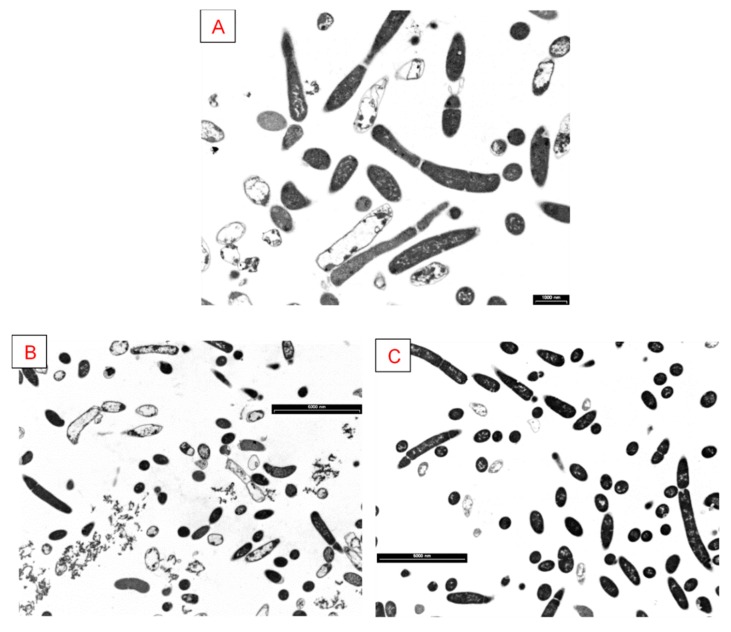
TEM images of *G. stearothermophilus* GF16: after overnight growth (**A**), in the presence of As(V) (**B**) and Cd(II) (**C**) at their respective MIC values. Scale: 1000 nm in **A**, elsewhere 6000 nm.

**Table 1 ijerph-17-02678-t001:** *G. stearothermophilus* resistance to heavy-metal ions.

Metal Ions	mM	±
As(III)	1.90	0.10
As(V)	117	3.00
Cd(II)	0.90	0.10
Co(II)	2.00	0.50
Co(III)	2.75	0.25
Cr(VI)	0.25	0.01
Cu(II)	4.10	0.10
Hg(II)	0.02	0.00
Ni(II)	1.30	0.10
V(V)	128	2.00

**Table 2 ijerph-17-02678-t002:** List of *G. stearothermophilus* strains with sequenced genomes, as reported in the National Center for Biotechnology Information (NCBI) genome databank.

Organism	Strain	Origin	Genome Size (Mb)	CG%	Gene	Protein	BioProject
*Geobacillus stearothermophilus*	10	Yellowstone thermal spring	3.67	52.61	3645	3312	PRJNA252389
*Geobacillus stearothermophilus*	DSM458	Austria sugar beet factory	3.46	52.10	3683	3165	PRJNA327158
*Geobacillus stearothermophilus*	B5	China rice stack	3.39	52.50	3426	3045	PRJNA513473

**Table 3 ijerph-17-02678-t003:** List of putative genes for As and Cd(II) resistance in *G. stearothermophilus* strains.

	*Geobacillus stearothermophilus* 10 (PRJNA252389)	*Geobacillus stearothermophilus* DSM458 (PRJNA327158)	*Geobacillus stearothermophilus* B5 (PRJNA513473)
Putative Proteins	***Locus***	***Locus***	***Locus***
ArsR	GT50_RS07590		EPB69_RS07030
			EPB69_RS15665
			EPB69_RS15730
ArsB	GT50_RS07510	GS458_RS16835	EPB69_RS15660
ArsC	GT50_RS07505	GS458_RS16830	EPB69_RS15655
	GT50_RS06280	GS458_RS15800	
CadA	GT50_RS12470	GS458_RS03700	EPB69_RS03440
CadC	GT50_RS12465	GS458_RS03695	EPB69_RS03435

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
