# Peer review of "Identification of a New Heavy-Metal-Resistant Strain of *Geobacillus stearothermophilus* Isolated from a Hydrothermally Active Volcanic Area in Southern Italy"

_ijerph, 2020, doi:10.3390/ijerph17082678_

Round 1

Reviewer 1 Report

This study aims to characterize new thermophilic heavy metal resistant microorganisms. The authors isolated a novel strain of the thermophilic bacterium Geobacillus stearothermophilus from solfataric terrains. The results of the MIC testing may deepen the readers’ understanding towards antibiotic and heavy metal resistance. However, there are several flaws need to be filled before publication.

For the determination of Minimal Inhibitory Concentrations (MIC) towards antibiotics, a frozen 111 (-80°C) stock of G. stearothermophilus was streaked on a LB/agar plate and incubated at 60°C 112 overnight. Is it enough to select LB/agar plate only for the MIC detection since microorganisms may show different MIC in different medium or growth conditions?

Since the genomes of the isolated pure culture has been extracted, profiling the resistant genes are essential and necessary for understanding the resistant mechanisms in this study.

“16SrRNA” must be modified to “16S rRNA” in the abstract.

Line 91: 16S rDNA should be modified to “16S rRNA”

The text should be polished carefully to exclude typos, such as line 270, the sentence should be modified as "in the presence of ..."; Line 175 Fe2O3; line 177 SO4-2

Author Response

Point-by-point response to reviewers

Reviewer 1 

Introduction has been improved and language edit applied as from indications in the Report Form

For the determination of Minimal Inhibitory Concentrations (MIC) towards antibiotics, a frozen 111 (-80°C) stock of G. stearothermophilus was streaked on a LB/agar plate and incubated at 60°C 112 overnight. Is it enough to select LB/agar plate only for the MIC detection since microorganisms may show different MIC in different medium or growth conditions?

R MIC determination towards both antibiotics and metal ions cells was obtained growing cells in liquid medium and following protocols already described in the Manual of Antimicrobial Susceptibility Testing (Rankin, 2005) and in our previously published papers. To better clarify the experimental conditions that we employed to measure the MIC reported in the Table 1, the paragraph has been rewritten adding more details.

Since the genomes of the isolated pure culture has been extracted, profiling the resistant genes are essential and necessary for understanding the resistant mechanisms in this study.

R In the present manuscript we identified the new strain of G. stearothermophilus through different molecular strategies but did not perform genome sequencing. Hopefully, the identification and characterization of resistance genes will be object of future studies. 

“16SrRNA” must be modified to “16S rRNA” in the abstract.

R Done, thank you.

Line 91: 16S rDNA should be modified to “16S rRNA”

R The sentence has been modified and the change applied.

The text should be polished carefully to exclude typos, such as line 270, the sentence should be modified as "in the presence of ..."; Line 175 Fe2O3; line 177 SO4-2

R All the mistakes have been corrected. We thank the reviewer for noticing.

Reviewer 2 Report

The authors isolated and identified a new strain that is tolerant to heavy metals. Chemical analysis and bioinformatics analyses were conducted. Overall the results seem reasonable for the newly identified strain.

It is recommended that the author further explain details on "peculiar taxonomy of Geobacillus" and discuss why 16S rRNA sequencing is not possible. It would be helpful to build a phylogenetic tree of new strain with existing identified strains to see the taxonomic relationships among similar species.

Line 226, "Figure 4" should be "Figure 3".

Shouldn't a control of LB agar with the new strain without As(V) be included? I also don't quite get the idea that a color change in Figure 3A suggesting that the new strain is incapable to extrude As(III). First, was As(III) added to the solution? Second, is this method precise enough to differentiate differences of As concentrations between As(III) and As(V)? A more accurate method, such as ICP, can be used to detect As in cells and in agar to evaluate the actual concentrations of As species and distribution in cells.

Please add metabolic (redox) pathways of relevant heavy metals (As and Cd) in the paper.

"Specie" has been used several times and it should be "species".

Author Response

Point-by-point response to reviewers

Reviewer 2

The methods have been described with additional details and conclusions and language have been improved as indicated in the Report Form

It is recommended that the author further explain details on "peculiar taxonomy of Geobacillus" and discuss why 16S rRNA sequencing is not possible. It would be helpful to build a phylogenetic tree of new strain with existing identified strains to see the taxonomic relationships among similar species.

R 16S rRNA sequencing has been performed successfully and in combination with MALDI-TOF-MS analysis allowed species identification; this has been better explained in the revised manuscript (see paragraph Isolation and identification of Geobacillus stearothermophilus both in Materials and Methods and Results and Discussion). Moreover, following the reviewer ‘s suggestion, in the new version of the manuscript we commented in more detail the peculiar taxonomy of Geobacillus and built a phylogenetic tree selecting related species. The phylogenetic tree is shown in the Figure 2 of the new manuscript.  We also submitted the 16S rRNA sequence to GenBank and report the accession number in the new version of the manuscript.

Line 226, "Figure 4" should be "Figure 3".

R Done, thank you.

Shouldn't a control of LB agar with the new strain without As(V) be included? I also don't quite get the idea that a color change in Figure 3A suggesting that the new strain is incapable to extrude As(III). First, was As(III) added to the solution? Second, is this method precise enough to differentiate differences of As concentrations between As(III) and As(V)? A more accurate method, such as ICP, can be used to detect As in cells and in agar to evaluate the actual concentrations of As species and distribution in cells.

R In the new version of the manuscript, we included more controls in the Figure 3. Moreover, the experimental conditions and the procedure have been rewritten and described in more detail. The assay has been performed only adding As(V) in order to evaluate As(III) formation by potential intracellular enzymatic activity and subsequent efflux; this has also been better explained in the new manuscript. Nevertheless, the experiment remains qualitative and preliminary and accordingly to the reviewer’s concern, the need to employ more accurate methods has also been discussed and will be addressed in a future work.

Please add metabolic (redox) pathways of relevant heavy metals (As and Cd) in the paper.

R The mechanisms of As and Cd resistance within the Geobacillus genus have been better discussed in the paragraph “Metal ion resistance and antibiotic susceptibility in G. stearothermophilus” of the new manuscript.  

"Specie" has been used several times and it should be "species".

R All the mistakes have been corrected. We thank the reviewer for noticing.

Reviewer 3 Report

The presented manuscript is very interesting. It is well written and easy to understand. It could be worth thinking about the metagenomic analysis of this strain and comparison with available sequences.

All small mistakes and doubts are highlighted in the text:

Line 13: "which"

Line 19: "and characterized by..."

Line 21: "of them"

Lines 49-50: "for"

Lines 53-55: Write it as a new sentence

Line 57: insufficient coma

Line 91: please describe the method of extraction

Line 92: please describe the conditions of PCR

Line 93: which databases

Line 96: "were"

Line 303: please pay attention on reference format to be the same in all positions

Author Response

Point-by-point response to reviewers

Reviewer 3

The methods have been described with additional details and language has been changed as from indications in the Report Form and in the annotated pdf.

All small mistakes and doubts are highlighted in the text:

Line 13: "which"

Line 19: "and characterized by..."

Line 21: "of them"

Lines 49-50: "for"

R Everything has been corrected, thank you.

Lines 53-55: Write it as a new sentence

R Accordingly to the reviewer suggestion, the phrase has been rewritten.

Line 57: insufficient coma

R We corrected the phrase accordingly.

Line 91: please describe the method of extraction

Line 92: please describe the conditions of PCR

R DNA extraction and PCR were commissioned to Eurofins genomics, this has been explicated in the new manuscript.

Line 93: which databases

R The database is NCBI (www.ncbi.nlm.nih.gov) and has been specified in the new manuscript.

Line 96: "were"

R We corrected accordingly.

Line 303: please pay attention on reference format to be the same in all positions

R All the reference section has been reviewed, thank you for the noticing.

Reviewer 4 Report

This manuscript describes a new strain of Geobacillus stearothermophilus that was isolated from a hot volcanic area in Italy. This strain is resistant to several heavy metals including As. Which appears to be consistent with other strains of this species. This is a well written manuscript and I have very few suggestions for improvement.

It would be helpful to include a phylogenetic tree that shows this strain within the Geobacillus genus. This would also help demonstrate the difficulty in assigning to a species based on 16S rRNA data alone.

While there are some comparisons with the other strains, and the genomic comparison with the other strains is a plus, I was left wondering if there is any information on MIC or resistance to heavy metals from the other strains other than the genomic potential. Adding the physiological attributes of these other strains and how that compares to this strain would help flush out the discussion.  

Along these same lines consider modifying Table 3 for an easier comparison between the three genomes. Something more similar to Table 1 would be easier to quickly make those type of comparisons.

Minor comments:

Line 74: awkward wording with “various erupting”

Line 77: remove “to sampling”

Line 92: Please provide details on the primers used.

Line 93: Please provide more details on how the sequences were analyzed (change spelling of analysed) and what databases were used.

Line 95: change to “Since the taxonomic identification at the species level could not be determined with 16S rRNA gene data, … “

Line 148: Change to “phosphate buffered saline”

Line 175-177: Verify some chemical compounds need subscripts.

Line 284: awkward wording.

Author Response

Point-by-point response to reviewers

Reviewer 4

The methods have been described with additional details and language changes applied as from indications in the Report Form.

It would be helpful to include a phylogenetic tree that shows this strain within the Geobacillus genus. This would also help demonstrate the difficulty in assigning to a species based on 16S rRNA data alone.

R Accordingly to the reviewer suggestion, in the new version of the manuscript we inserted a phylogenetic tree (Figure 2). Moreover, in in the new manuscript we commented in more detail the peculiar taxonomy of Geobacillus. We also submitted the 16S rRNA sequence to GenBank and report the accession number in the new version of the manuscript.

While there are some comparisons with the other strains, and the genomic comparison with the other strains is a plus, I was left wondering if there is any information on MIC or resistance to heavy metals from the other strains other than the genomic potential. Adding the physiological attributes of these other strains and how that compares to this strain would help flush out the discussion.  

R Thanks to the reviewer observation the paragraph “Metal ion resistance and antibiotic susceptibility in G. stearothermophilus” has been rewritten in order to contextualize the physiological features of the novel isolate in comparison to other characterized species.

Along these same lines consider modifying Table 3 for an easier comparison between the three genomes. Something more similar to Table 1 would be easier to quickly make those type of comparisons.

R Table 3 has been modified following reviewer’s suggestion.

Minor comments:

Line 74: awkward wording with “various erupting”

R The sentence has been modified in: “….one of the various eruptive vents generated within the wide volcanic field of the Campi Flegrei since ca. 4100 years [42]”.

Line 77: remove “to sampling”

R Done, thank you.

Line 92: Please provide details on the primers used.

Line 93: Please provide more details on how the sequences were analyzed (change spelling of analysed) and what databases were used.

R The paragraph has been modified accordingly.

Line 95: change to “Since the taxonomic identification at the species level could not be determined with 16S rRNA gene data, … “

R We corrected the sentence accordingly.

Line 148: Change to “phosphate buffered saline”

R Corrected, thank you.

Line 175-177: Verify some chemical compounds need subscripts.

R Corrected, thank you.

Line 284: awkward wording.

R Accordingly, the phrase has been rewritten as follows: “We succeeded in isolating a microorganism with an optimal growth temperature of 60°C from a water-poor mud”. 

Round 2

Reviewer 1 Report

I think the manuscript has been significantly improved and now warrants publication in the IJERPH.

Reviewer 2 Report

The revised version has addressed my comments.